# Purkinje cell outputs selectively inhibit a subset of unipolar brush cells in the input layer of the cerebellar cortex

Chong Guo[†], Stephanie Rudolph[‡], Morgan E Neuwirth, Wade G Regehr*

Department of Neurobiology, Harvard Medical School, Boston, United States

**Abstract** Circuitry of the cerebellar cortex is regionally and functionally specialized. Unipolar brush cells (UBCs), and Purkinje cell (PC) synapses made by axon collaterals in the granular layer, are both enriched in areas that control balance and eye movement. Here, we find a link between these specializations in mice: PCs preferentially inhibit metabotropic glutamate receptor type 1 (mGluR1)-expressing UBCs that respond to mossy fiber (MF) inputs with long lasting increases in firing, but PCs do not inhibit mGluR1-lacking UBCs. PCs inhibit about 29% of mGluR1-expressing UBCs by activating GABA$_A$ receptors (GABA$_A$Rs) and inhibit almost all mGluR1-expressing UBCs by activating GABA$_B$ receptors (GABA$_B$Rs). PC to UBC synapses allow PC output to regulate the input layer of the cerebellar cortex in diverse ways. Based on optogenetic studies and a small number of paired recordings, GABA$_A$R-mediated feedback is fast and unreliable. GABA$_B$R-mediated inhibition is slower and is sufficiently large to strongly influence the input-output transformations of mGluR1-expressing UBCs.

**\*For correspondence:**
wade_regehr@hms.harvard.edu

**Present address:** [†]Department of Brain and Cognitive Sciences and McGovern Institute for Brain Research, Massachusetts Institute of Technology, Cambridge, United States; [‡]Albert Einstein College of Medicine, New York, United States

**Competing interests:** The authors declare that no competing interests exist.

## Introduction

Although different lobules of the cerebellar cortex are engaged in diverse motor and non-motor behaviors (*Ito, 1998*; *Kim and Thompson, 1997*; *Massion, 1992*; *Murdoch, 2010*; *Schmahmann and Sherman, 1998*; *Siuda et al., 2014*; *Van Overwalle et al., 2014*; *Villanueva, 2012*), a basic circuit motif is repeated within each functional compartment. Mossy fibers provide excitatory inputs to granule cells (GrCs) (*Delvendahl and Hallermann, 2016*; *DiGregorio et al., 2002*; *Huang et al., 2013*; *Sotelo, 2008*), which in turn excite Purkinje cells (PCs) that provide the output of the cerebellar cortex. In addition, Golgi cells (GoCs) inhibit GrCs, and molecular layer interneurons inhibit PCs (*Eccles, 2013*). Regional synaptic and cellular specializations exist beyond this basic circuit motif, presumably to meet the computational demands associated with specific behaviors (*Cerminara et al., 2015*; *Diño et al., 1999*; *Guo et al., 2016*; *Kozareva et al., 2021*; *Sekerková et al., 2014*; *Suvrathan et al., 2016*).

One of the most obvious specializations is that the density of unipolar brush cells (UBCs) exhibits large regional variation (*Diño et al., 1999*; *Sekerková et al., 2014*). UBCs are excitatory interneurons located in the granular layer that are usually innervated by a single MF (*Mugnaini and Floris, 1994*). UBCs contribute to temporal processing by converting short-lived MF signals into long-lasting changes in UBC activity (*Kennedy et al., 2014*; *Kinney et al., 1997*; *Kreko-Pierce et al., 2020*; *Locatelli et al., 2013*; *Mugnaini and Floris, 1994*; *Mugnaini et al., 2011*; *Rossi et al., 1995*; *van Dorp and De Zeeuw, 2014*). MFs excite UBCs that express metabotropic glutamate receptor type I (mGluR1), suppress firing in UBCs where mGluR2 is prominent, and evoke more complex responses in other UBCs that express both mGluR1 and mGluR2 (*Borges-Merjane and Trussell, 2015*; *Guo et al., 2021*; *Kinney et al., 1997*; *Knoflach and Kemp, 1998*; *Rossi et al., 1995*; *Russo et al., 2008*; *van Dorp and De Zeeuw, 2014*; *Zampini et al., 2016*). UBCs are often disregarded in descriptions of the circuitry of the cerebellar cortex. However, UBCs are present in all

areas and are exceptionally dense in some regions, notably those involved in vestibular function (*Diño et al., 1999*; *Takács et al., 1999*). UBCs are more prevalent and widely distributed in larger mammals including humans (*Diño et al., 1999*; *Munoz, 1990*), suggesting a special role in the complex cerebellar computations relating to higher cognition.

PC feedback via collaterals that are restricted to parasagittal planes is another noteworthy regional specialization (*Witter et al., 2016*). In addition to sending an axon to the deep cerebellar nuclei, each PC axon has a collateral that inhibits PCs (*Bernard and Axelrad, 1993*; *Bernard et al., 1993*; *Bornschein et al., 2013*; *de Solages et al., 2008*; *Orduz and Llano, 2007*; *Watt et al., 2009*; *Witter et al., 2016*) and several types of inhibitory interneurons in the cerebellar cortex (*Crook et al., 2007*; *Hirono et al., 2012*; *Witter et al., 2016*). In some regions, PCs make extensive contacts within the granular layer and inhibit GrCs (*Guo et al., 2016*). This feedback is mediated primarily by GABA$_A$ receptors (GABA$_A$Rs) and has a prominent slow component that suggests the involvement of the extrasynaptic high-affinity GABA$_A$Rs. These PC to GrC synapses allow the output of the cerebellar cortex to provide slow negative feedback to inhibit the input layer. Like UBCs, these synapses are also most prominent in regions involved in vestibular function.

The regional overlap of high densities of UBCs and granular layer PC synapses raises the possibility that PCs might inhibit UBCs and thereby allow PC feedback to refine temporal processing. Here, we examine whether PCs inhibit UBCs. We find that PCs phasically inhibit a subset of UBCs by activating GABA$_A$Rs, and that UBCs with prominent mGluR1 signaling are preferentially targeted. PCs also provide long-lasting inhibition to most mGluR1-expressing UBCs by activating GABA$_B$ receptors (GABA$_B$Rs). In these ways, PC inhibition of mGluR1-expressing UBCs provides direct feedback from the output layer to input layer to influence the first stage of cerebellar processing.

## Results

### Regional distribution of PC collaterals and mGluR1$^+$ UBCs

High-power confocal imaging of vermal and floccular slices was used to assess the regional density of PC collateral synapses and UBCs, and to detect putative synaptic connections between PCs and UBCs. PC presynaptic boutons were labeled in Pcp2-Cre×synaptophysin-tdTomato mice (n=2). Weaker labeling was evident in PC dendrites and somata (red, *Figure 1A*). Inhibitory synapses were labeled with vesicular GABA transporter (VGAT) immunofluorescence (green, *Figure 1A*). PC synapses were automatically identified with a custom deep neural network using VGAT and synaptophysin-tdTomato immunofluorescence (*Figure 1—figure supplement 1*, see Materials and methods). For visualization, annotated PC synapses were colored in cyan, and the PC and molecular layers are shown in gray based on the synaptophysin-tdTomato signal (*Figure 1B*). In the same slices, mGluR1 immunofluorescence labeled UBCs and PCs. mGluR1 labeling of UBC brushes is intense and distinctive (*Borges-Merjane and Trussell, 2015*; *McDonough et al., 2020*; *Nunzi et al., 2002*). In our images, mGluR1 labeling of UBCs dominates the signals in both the high and low magnification of the granular layer. For display purposes, mGluR1 labeling in the PC and molecular layers was masked to highlight UBCs in the granular layer (magenta, *Figure 1C-K*).

The densities of UBCs and granular layer PC synapses exhibited considerable regional variability. The highest densities of both UBCs and granular layer PC synapses were found in vestibular regions (lobules IX, X, FL, and PFL; see *Figure 1C, D and F-I*) and in the oculomotor vermis (lobules VIb and VII, *Figure 1C*). Much lower densities were observed in the anterior cerebellar cortex (lobules I–V and VIa; see *Figure 1C,E*) and lobule VIII. In lobule X, there was a particularly high density of PC collateral synapses and UBCs (*Figure 1J*). High-power images of lobule X suggest that PC synapses directly terminate on the brushes and the cell bodies of many UBCs (*Figure 1K*).

### PC collaterals provide GABA$_A$R-mediated inhibition to mGluR1$^+$ UBCs

To determine whether PCs directly inhibit UBCs, we optically stimulated PC axons in Pcp2-Cre×ChR2 mice while recording from UBCs (*Figure 2A*). We performed these experiments in lobule X of acute slices in the presence of AMPA, NMDA, glycine, and GABA$_B$R blockers and a high chloride internal solution. UBCs exhibit diverse responses to MF activation, ranging from excitation mediated by AMPARs and mGluR1, to inhibition mediated by mGluR2, with intermediate cells having more diverse responses mediated by a combination of mGluR1 and mGluR2 (*Guo et al., 2021*).

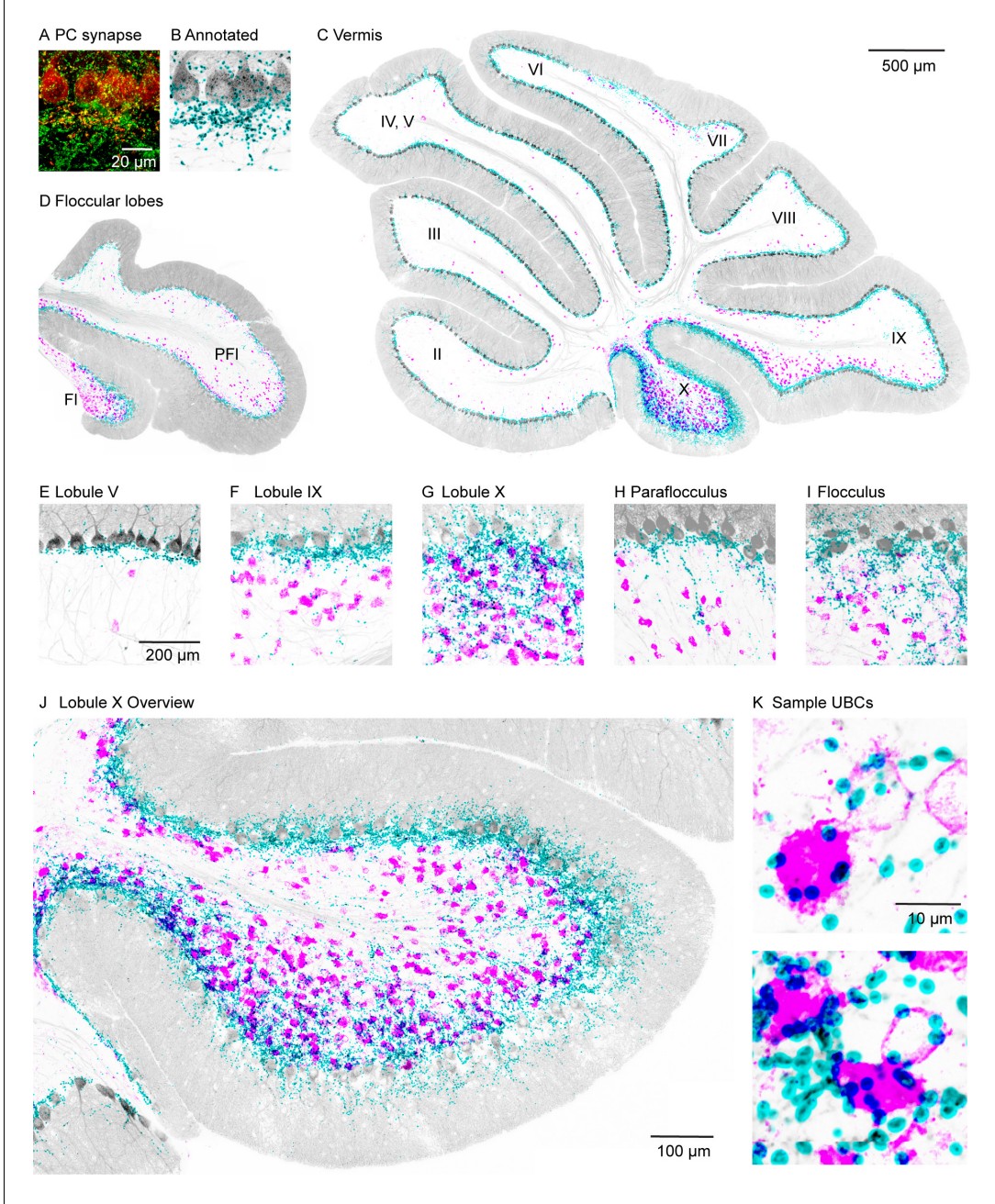

**Figure 1.** PC collateral synapses colocalized with mGluR1 UBCs in the vestibular lobules. (**A**) Maximal intensity projection of a 15-µm confocal z-stack images of PC collateral synapses co-labeled by synaptophysin-tdTomato (red) and VGAT (green). (**B**) Annotated PC collateral synapses (cyan) and the synaptophysin-tdTomato signal (gray). (**C**) A vermal slice of cerebellum analyzed as in (**B**) with synapses labeled in cyan, synaptophysin-tdTomato in gray, and the mGluR1 labeled UBCs dendritic brush in magenta. (**D**) Same analysis on a coronal slice of a floccular lobe. (**E–I**) Zoomed in view of selective lobules showing variable degrees of colocalization between PC synapses and UBCs. (**J**) Expanded view of lobule X. (**K**) Two-sample UBCs showing clear examples of collateral synapses onto the brush and the soma. mGluR1, metabotropic glutamate receptor type 1; PC, Purkinje cell; UBC, unipolar brush cell; VGAT, vesicular GABA transporter.

The online version of this article includes the following figure supplement(s) for figure 1:

**Figure supplement 1.** Automatic synapse detection using a convolutional neural network.

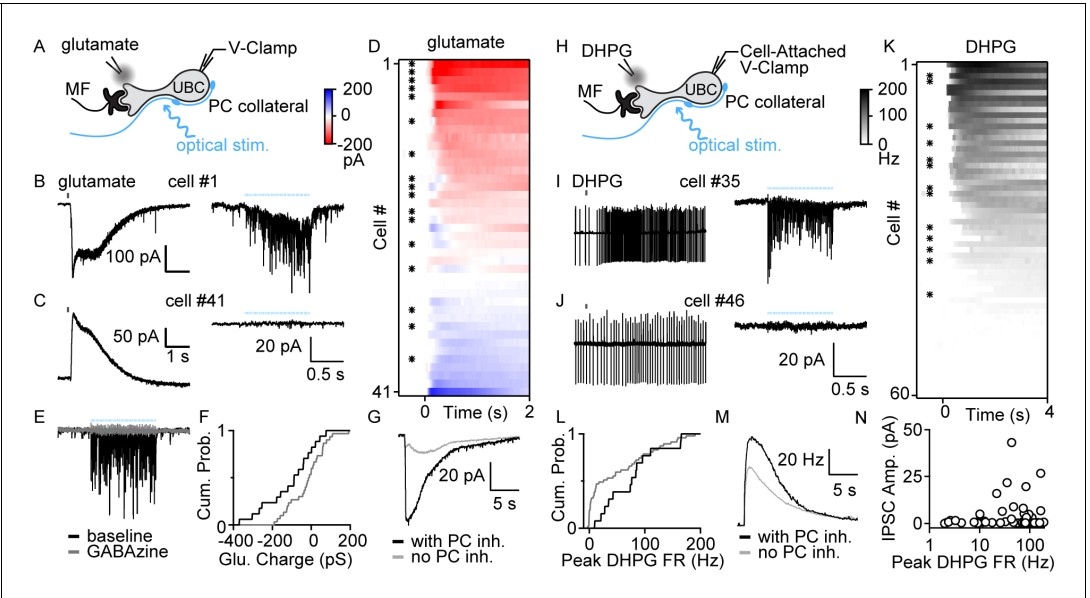

**Figure 2.** PC collaterals preferentially inhibit mGluR1$^+$ UBCs via fast GABA$_A$R-mediated feedback. (A) Schematic showing a mossy fiber (MF) and a PC collateral innervating a UBC. A whole-cell electrode was used to voltage clamp the UBC, and responses were measure for either a glutamate puff applied with a nearby electrode or to optical stimulation of PC collaterals. (B) Whole-cell recordings of glutamate-evoked currents (left) and to optically evoked PC inhibition (right) are shown for cell #1 of (D). (C) As in (B), but for cell #41 of (D). (D) Summary of cells in which responses to glutamate puffs were measured, with red corresponding to an excitatory inward current and blue corresponding to an inhibitory outward current. Cells with light-evoked inhibitory synaptic current are indicated with a black dot (n=41). (E) Light-evoked synaptic currents were recorded in baseline (black) and after the addition of the GABA$_A$ receptor antagonist gabazine (gray). (F) Normalized cumulative distributions of charge evoked by a glutamate puff for cells that either had light-evoked responses (black) or that did not (gray). (G) Average currents evoked by a glutamate puff for cells that had PC feedback (black) and that did not have PC feedback (gray). (H) Schematic as in (A), but for application of the mGluR1 agonist DHPG. DHPG-evoked increases in firing were measured with a cell-attached electrode, and then optically evoked synaptic currents were measured with a whole-cell voltage-clamp recording. (I) DHPG-evoked increases in firing (left) and light-evoked synaptic currents (right) are shown for cell #35 in (K). (J) Same as (I) but for cell #45 in (K). (K) Summary of cells in which firing evoked by DHPG puffs was quantified. Cells with light-evoked inhibitory synaptic current are indicated with a black dot (n=60). (L) Normalized cumulative distributions of charge evoked by a DHPG puff for cells that either had light-evoked responses (black) or that did not (gray). (M) Average currents evoked by a DHPG puff for cells that had PC feedback (black) and that did not have PC feedback (gray). (N) Scatter plot of IPSC amplitudes as a function of the peak firing rates for the cells in (K). DHPG, (S)-3,5-Dihydroxyphenylglycine; IPSC, inhibitory post-synaptic current; mGluR1, metabotropic glutamate receptor type 1; PC, Purkinje cell; VGAT, vesicular GABA transporter.

The online version of this article includes the following figure supplement(s) for figure 2:

**Figure supplement 1.** GABA$_A$ and GABA$_B$ receptor subunits are preferentially expressed in mGluR1-positive UBCs.

Electrophysiological studies have shown that fast inhibition in UBCs is mediated by a combination of glycine and GABA$_A$Rs (*Dugué et al., 2005*; *Rousseau et al., 2012*), but that mGluR1-lacking UBCs lack GABA$_A$Rs (*Rousseau et al., 2012*). Consistent with these observations, molecular profiling of UBCs using single-cell RNA sequencing (scRNASeq) (*Kozareva et al., 2021*) indicate that α1, β2, β3, and γ2 subunits of GABA$_A$Rs have a gradient of expression levels and are selectively enriched in mGluR1-positive UBCs (*Figure 2—figure supplement 1B*), but that glycine receptors are broadly expressed by UBCs (*Figure 2—figure supplement 1C*). This opens the possibility that the molecular identity of UBCs could determine whether they receive PC feedback. We therefore characterized UBC responses to brief puffs of glutamate (*Figure 2B,C* left) prior to measuring optically evoked synaptic responses (*Figure 2B,C* right). Glutamate (1mM, 50ms) evoked a continuous range of metabotropic responses across UBCs with variable degrees of excitation versus inhibition, as is apparent in the sorted heatmap of glutamate-evoked response (*Figure 2D*). Optical stimulation evoked currents in a subpopulation of these cells (*Figure 2B* right, *Figure 2D* asterisk marks, 17/41 cells) but not in others (*Figure 2C*, right, 24/41 cells). GABA$_A$R antagonist gabazine eliminated these light-evoked synaptic currents (*Figure 2E*). We compared the average responses to a glutamate puff for UBCs that had light-evoked GABA$_A$ currents with those that did not. Cumulative histograms for glutamate-evoked charge transfer (*Figure 2F*), and the average glutamate-evoked current for UBCs

with or without PC feedback (*Figure 2G*), show that PC GABA$_A$ feedback preferentially targets UBCs that are excited by glutamatergic inputs.

To further characterize the properties of UBCs that receive PC feedback, we categorized UBCs based on their responses to the mGluR1 agonist (S)-3,5-Dihydroxyphenylglycine (DHPG) prior to characterizing light-evoked synaptic responses (*Figure 2H*). DHPG (100mM, 50 ms) puffs evoked spiking in a subset of UBCs (*Figure 2I,J*, left, K 45/60). After recording the DHPG-evoked increase in firing with a cell-attached electrode, we obtained a whole-cell configuration and measured light-evoked responses. 13/45 DHPG-responsive UBCs also had significant light-evoked responses (*Figure 2I,J*, right, K). We then grouped UBCs by the presence (*Figure 2I*, right and asterisk in K) or absence of PC feedback (*Figure 2J*, right). Cumulative DHPG-evoked peak firing in UBCs with PC feedback is shifted toward greater response than those without feedback (*Figure 2L*). Furthermore, the average DHPG-evoked instantaneous firing in UBCs with PC feedback is also slightly bigger than those without (*Figure 2M*). Cumulative histograms for DHPG-evoked firing rate increases in UBCs (*Figure 2L*), and the average DHPG-evoked firing rate increases for UBCs with or without PC feedback (*Figure 2M*), show that PC GABA$_A$R-mediated feedback preferentially targets UBCs with larger mGluR1-mediated excitation. There is considerable scatter in the amplitudes of the average GABA$_A$ responses evoked by optical stimulation (*Figure 2N*).

We also directly recorded from connected PC-UBC pairs (*Figure 3*, n=2). An on-cell recording allowed us to noninvasively monitor PC spiking while recording synaptic responses in a UBC. Many UBC inhibitory post-synaptic currents (IPSC)s were timed to PC spikes (*Figure 3A*, top), as is readily appreciated in spike-triggered averages of UBC IPSCs (*Figure 3B*). The IPSC latency was 1.5±0.4 ms (*Figure 3C,D,F*) and the distribution of response amplitude was approximately Gaussian (*Figure 3E*). While the response potency is large (*Figure 3F*), there was a high failure rate of approximately 85% (*Figure 3F*).

## GABA$_B$R-mediated slow PC inhibition in mGluR1$^+$ UBCs

One of our goals was to determine if PCs also activate GABA$_B$Rs and thereby tonically inhibit UBCs. Previous studies established that only mGluR1-positive UBCs contain GABA$_B$Rs (*Kim et al., 2012*). This is consistent with scRNA-Seq analysis of UBCs (*Figure 2—figure supplement 1A and D*). GABA$_B$Rs comprises GB1 and GB2 subunits (*Gassmann and Bettler, 2012*) that are encoded by *Gabbr1* and *Gabbr2*. UBCs exhibit a gradient of expression of *Gabbr2* that is similar to *Grm1* (mGluR1), and *Gabbr1* has a less pronounced gradient of expression (*Figure 2—figure supplement 1D*). The differences in the expression patterns of *Gabbr2* and *Gabbr1* are intriguing, because it is thought that GABA$_B$Rs are heterodimers consisting of GABA$_{B1}$ and GABA$_{B2}$ receptors (*Frangaj and Fan, 2018*). Nonetheless, these findings are

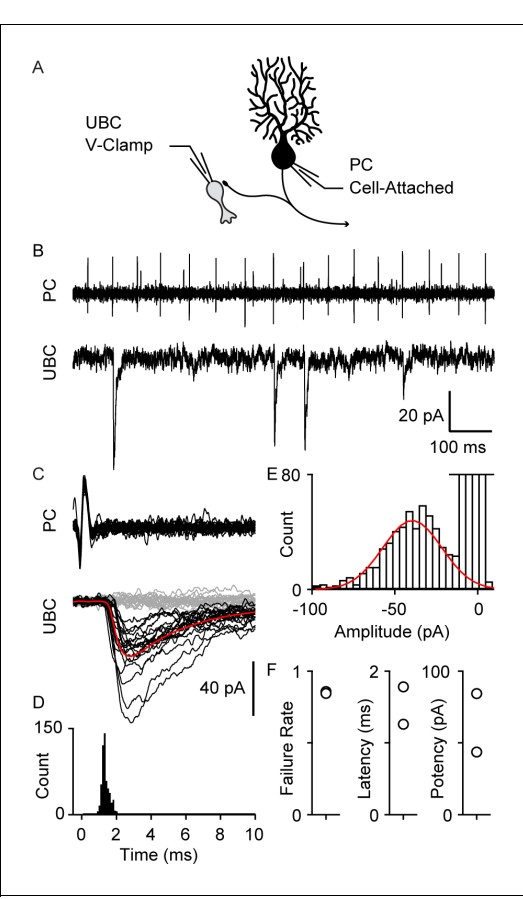

**Figure 3.** Paired recordings of PC to UBC synaptic connections. (**A**) Schematic of the paired recording configuration. (**B**) A representative pair showing a cell-attached PC recording (top) and whole-cell voltage-clamp recording of a synaptically connected UBC (bottom). (**C**) Same two cells showing time-aligned PC spikes (top) and associated IPSCs in UBCs (bottom), with successes in black, failures in gray, and average response from success trials in red. (**D**) Histogram of IPSC latency (bin size=0.1 ms). (**E**) Histogram of IPSC amplitude (bin size=4 pA) and Gaussian fit over success trials (red). (**F**) Failure rate (left), latency (middle), and potency (left) PC to UBC synapse (n=2 cells). IPSC, inhibitory post-synaptic current; PC, Purkinje cell; UBC, unipolar brush cell.

consistent with mGluR1$^+$ UBCs preferentially expressing GABA$_B$Rs, and we therefore restricted our recordings to cells that responded to DHPG. We found that in whole-cell recordings the currents evoked by GABA$_B$R agonists washed out within minutes (*Figure 4—figure supplement 1A and B*, gray). This washout was accompanied by an increase in the leak current (*Figure 4—figure supplement 1C*). In contrast, in perforated patch recordings, GABA$_B$ responses were large and stable, and leak currents were also stable (*Figure 4—figure supplement 1*, black). These experiments established that perforated patch recordings facilitate the measurement of PC-evoked GABA$_B$ responses.

We examined the PC-evoked GABA$_B$ responses in UBCs with an approach similar to that used to record GABA$_A$ responses, except that we blocked GABA$_A$Rs, used perforated patch recordings, and elevated PC firing for 2 s to activate the GABA$_B$Rs on a longer time scale (*Figure 4A*). We stimulated the entire lobule X with graded light levels to approximately double and triple PC firing rates (*Figure 4B–D*). Optical stimulation evoked outward currents in all UBCs (*Figure 4E–G*, 13/13). GABA$_B$R-mediated inhibition was slow (*Figure 4F*, rise time constant=583±6 ms, decay time constant=533±5 ms). The GABA$_B$R antagonist CGP eliminated these light-evoked responses (*Figure 4K*). The average inhibition of 3.8±2.2 pA was sufficiently large to strongly suppress or entirely shut down spontaneous UBC firing (*Figure 4H–J* in 9/10 spontaneous firing UBCs). These findings suggest that PC firing rates dynamically regulate the activation of GABA$_B$Rs, and by extension the spontaneous firing of most mGluR1$^+$ UBCs. As with the direct GABA$_A$R-mediated response, it is unclear whether PCs inhibit all mGluR1$^+$ UBCs, or if we are unable to optogenetically activate PC inputs onto a small fraction of UBCs for technical reasons, such as severed and unhealthy PC axons in our brain slice preparation. With this GABA$_B$R-dependent mechanism, PCs inhibit a much larger fraction of mGluR1$^+$ UBCs (90–100%, *Figure 4G*, 13/13 DHPG responding cells and *Figure 4J*, 9/10 spontaneous firing), compared to GABA$_A$R-dependent inhibition (29%, *Figure 2K*, 13/45 DHPG-responding cells).

## GABA$_B$R-mediated PC-UBC inhibition temporally sharpens UBC response

In addition to suppressing spontaneous activity, PC firing and GABA$_B$R activation could control the excitability of UBCs and their response to depolarization. To assess the effect of small tonic hyperpolarizing currents on UBC excitability, we examined how such currents influence UBC responses to current steps using perforated patch recordings. In spontaneously firing UBCs, small depolarizing current steps evoked firing that declined to a somewhat reduced steady-state response that was proportional to the current injected (*Figure 5A,B and C*, gray left). A 5-pA tonic hyperpolarizing current, drastically altered these responses, with current steps evoking a much larger transient response and reduced steady-state firing (*Figure 5A,B and C*, black right). A summary of average spiking responses to different current step amplitudes of all UBCs without and with tonic hyperpolarization is shown in *Figure 5B and C*. In all cells, hyperpolarization increased the difference between peak and steady-state firing rates (*Figure 5C*, n=4). We also directly examined the effects of PC firing on UBC excitability. In these experiments, we blocked GABA$_A$Rs to isolate the effect of GABA$_B$Rs on spiking. Full-field optical excitation of PCs activated GABA$_B$R-mediated inhibition and produced qualitatively similar effects on spiking as hyperpolarizing current injections (*Figure 5D*, blue, right). Optical activation of PC feedback through GABA$_B$R hyperpolarized UBCs (*Figure 5D*, blue, right), and current steps evoked larger initial responses and smaller steady-state responses (*Figure 5E,F*, blue vs. gray). These counterintuitive effects on spiking, in which a hyperpolarization increases initial responses, are consistent with previously described properties of UBCs (*Perez-Reyes, 2003*). They suggest that the effects of PC inhibition of UBCs may be more complex than simply suppressing spiking in UBCs.

MF activity elevates firing in the input layer of the cerebellar cortex that in turn alters PC spiking. The PC to UBC feedback described here has the potential to allow PC output to dynamically regulate the firing of mGluR1-expressing UBCs. We have previously shown that a brief MF burst can activate mGluR1 to evoke increases in firing that last for more than 10 s in some UBCs (*Guo et al., 2021*). We briefly applied DHPG to crudely mimic MF-evoked activation of mGluR1-expressing UBCs, and baclofen to mimic GABA$_B$R activation arising from elevated PC firing (*Figure 5G,J*). Brief application of baclofen following DHPG reduced the number of evoked spikes (*Figure 5H–J*). Thus, activation of GABA$_B$Rs, as could occur when PC activity is elevated, can strongly suppress mGluR1-activated UBC firing (see Discussion).

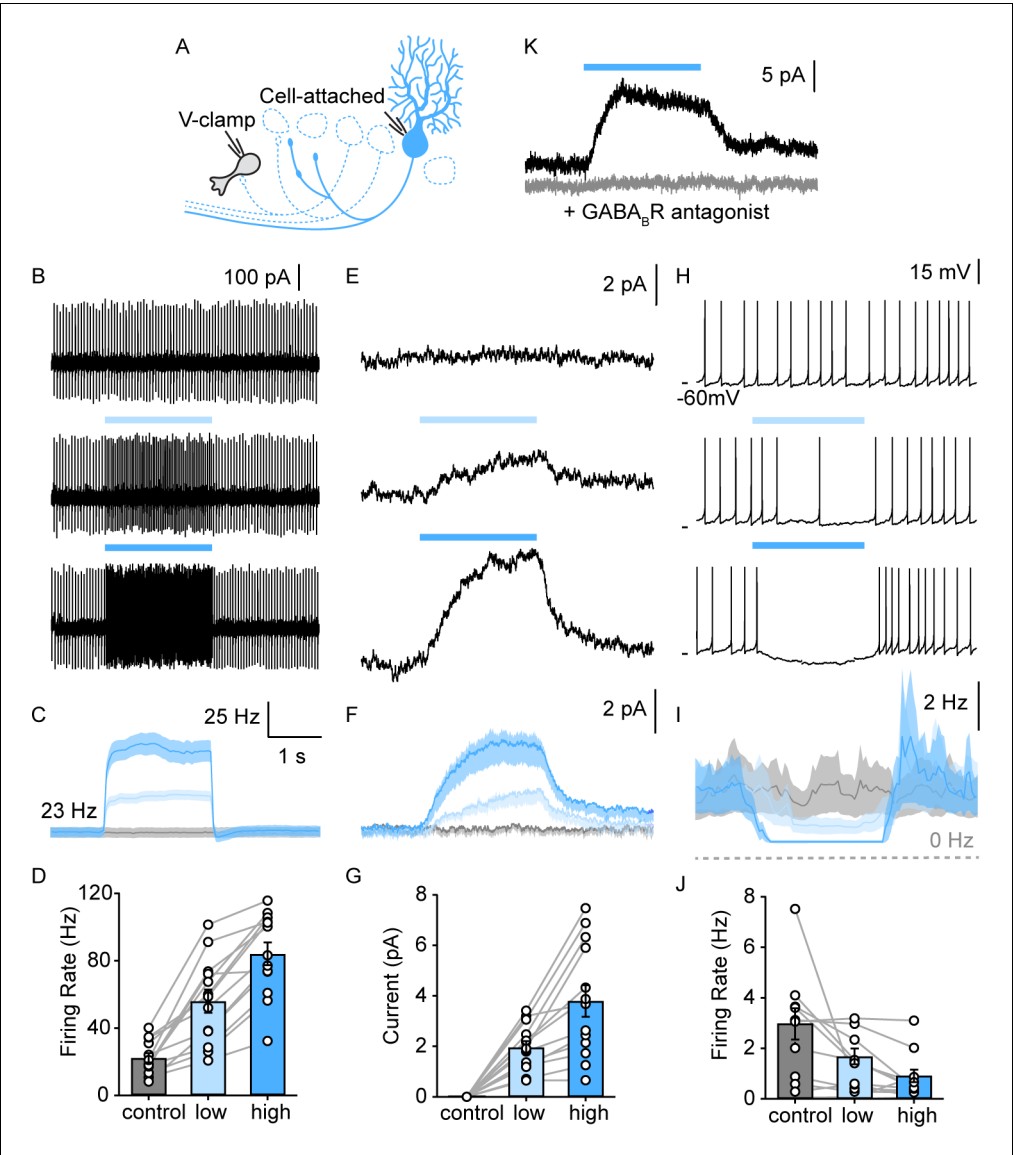

**Figure 4.** PC collaterals provide slow GABA$_B$ receptor-mediated inhibition onto lobule X UBCs. (**A**) Schematic of full-field optical stimulation for modulating PC firing rate in lobule X. (**B**) Cell-attached recording of spontaneous PC spikes for no (top), low intensity (middle), and high intensity (bottom) full-field optical stimulations. (**C**) Average instantaneous firing rate of PCs with no (gray), low (light blue), and high (blue) intensity stimulations (n=13). (**D**) Summary of PC firing rate with no (gray), low (light blue), and high (blue) intensity light-stimulations of PCs (n=13). (**E**) Voltage-clamp recordings in UBC revealed a slow inhibitory response that is modulated by PC firing rate, same optical stimulation conditions as in (**B**). (**F**) Average evoked current of UBCs during stimulations as in (**E**) (n=13). (**G**) Summary of inhibitory current amplitude in UBCs with no (gray), low (light blue), and high (blue) intensity light-stimulations of PCs (n=13). (**H**) Full-field activation of PCs suppressed the frequency of UBC firing. (**I**) Average instantaneous firing rate of UBCs with no (gray), low (light blue), and high (blue) intensity stimulations (n=13). (**J**) Summary of firing rate changes in spontaneous firing UBCs with no (gray), low (light blue), and high (blue) intensity light-stimulations of PCs (n=10). (**K**) PC evoked slow outward currents in UBC in the presence of GABA$_A$R blockers under voltage-clamp before (black) and after blocking GABA$_B$R (gray) (n=1). PC, Purkinje cell; UBC, unipolar brush cell.

The online version of this article includes the following figure supplement(s) for figure 4:

**Figure supplement 1.** Perforated patch recordings provide stable responses to a GABA$_B$-receptor agonist.

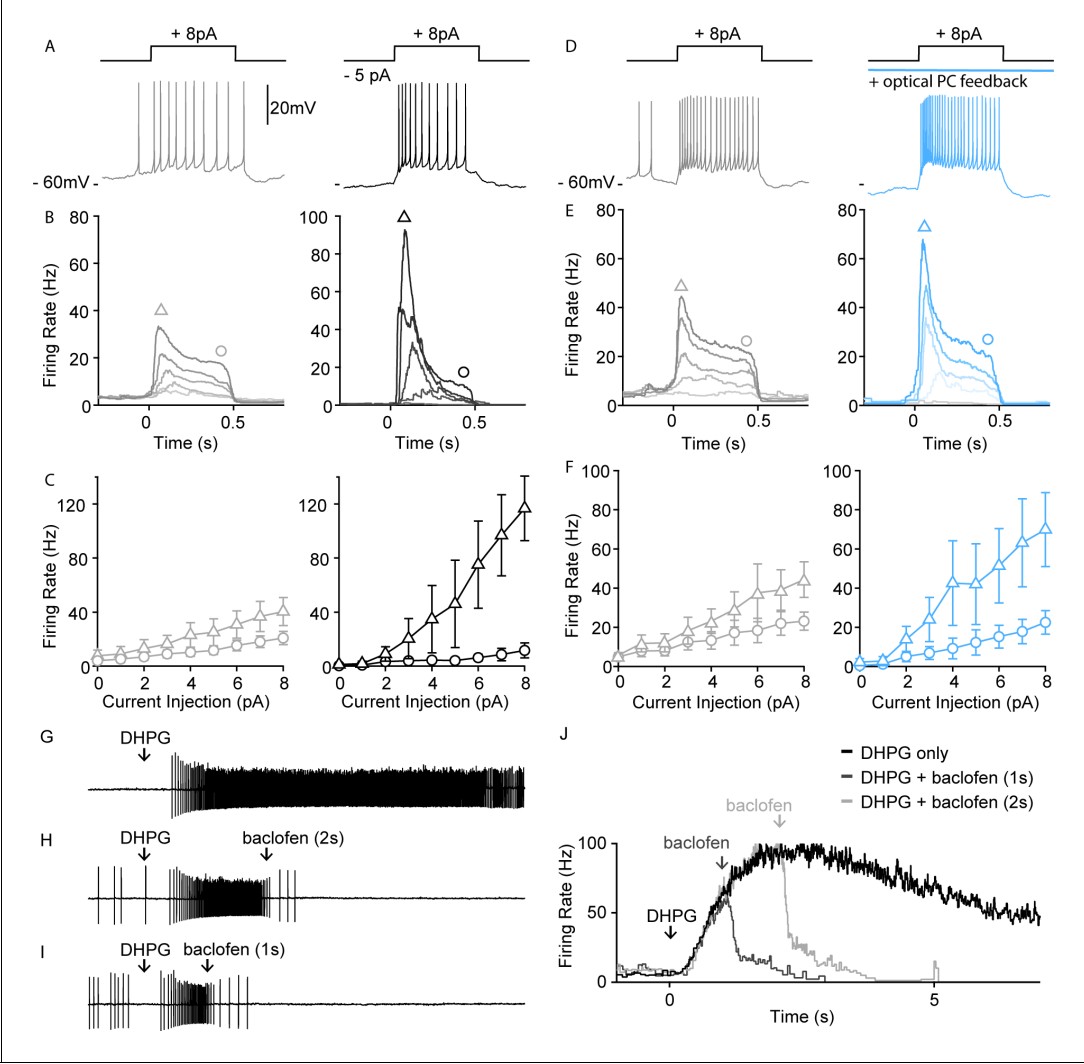

**Figure 5.** PC-UBC feedback via GABA$_B$R sharpens temporal response to current injections. (**A**) Sample perforated current-clamp recording of UBC spiking response to an 8 pA current step (500 ms), without (gray, left) and with a hyperpolarizing tonic current (black, right). (**B**) Average instantaneous firing rate to current steps of (0, 2, 4, 6, and 8 pA, from light to dark shade) without (gray, left) and with a hyperpolarizing tonic current (black, right) (n=4). (**C**) Summary of peak firing rate (triangle marker, mean ± SEM) and steady-state firing rate (circle marker, mean ± SEM) for the two conditions as in (B) (n=4). (**D**) Same summary plot as in (A) for control (gray, left) versus optical PC feedback (blue, right). (**E**) Same summary plot as in (B) for control (gray, left) versus optical PC feedback (blue, right) (n=5). (**F**) Same summary plot as in (C) for control (gray, left) versus optical PC feedback (blue, right) (n=5). (**G**) In a sample UBC pressure-application of a mGluR1 agonist (DHPG 100 µM, 10 ms) evoked persistent spiking for a few seconds. (**H**) Pressure-applied GABA$_B$R agonist (baclofen 250 µM, 100 ms) 2 s following mGluR1 agonist application readily reduced the firing rate of persistent spiking response. (**I**) Same as in (B) but only 1 s following mGluR1 agonist application. (**J**) Average instantaneous firing rate of the sample UBC under mGluR1 agonist application only (black), mGluR1$^+$ GABA$_B$R agonist applied 1 s apart (dark gray) or 2 s apart (light gray). PC, Purkinje cell; UBC, unipolar brush cell.

## Discussion

The PC to UBC synapses described here link two previously described regional specializations of the cerebellar cortex: the high densities of UBCs and the prevalence of PC synapses within the granular layer. PC to UBC synapses provide a new way for PC outputs to dynamically regulate the temporal transformations by UBC populations within the input layer.

### Target specificity of GABA$_A$ and GABA$_B$ PC to UBC feedback

PCs inhibit a subset of UBCs. Optical stimulation of PC terminals evoked GABA$_A$R-mediated responses in 29% (30/101) of UBCs. Light-evoked PC GABA$_A$R-mediated inhibition was much more

common in cells in which glutamate evoked a large excitatory current but was also observed in three cases where glutamate evoked inhibitory currents (*Figure 2D*). This may reflect the graded nature of metabotropic glutamate receptor signaling in UBCs in which outward mGluR1-mediated currents and inward mGluR2-mediated currents can be present in the same cell, and a large mGluR2 component could obscure a small mGluR1 component in some cells (*Guo et al., 2021*; *Kozareva et al., 2021*). When we used DHPG to identify mGluR1$^+$ cells, we found that PCs did not inhibit any mGluR1$^-$ UBCs (0/15), (*Figure 2K*). This is in keeping with the observations that mGluR1$^-$ UBCs do not respond to either GABA$_A$R or GABA$_B$R agonists (*Kim et al., 2012*; *Rousseau et al., 2012*), that they do not express GABA$_A$R or GABA$_B$R subunits (*Rousseau et al., 2012*; *Figure 2—figure supplement 1*), and that PCs do not release glycine (*Tanaka and Ezure, 2004*). In contrast to PCs that only inhibit a subset of UBCs, glycinergic GoCs can inhibit all UBCs, because ionotropic glycine receptors and glycinergic IPSCs are present in all UBCs (*Dugué et al., 2005*; *Rousseau et al., 2012*).

Our findings provide new insights into the previous immunohistochemical characterization of the inhibitory synapses onto UBCs that suggested approximately 50% co-release GABA and glycine and the rest only release GABA (*Dugué et al., 2005*; *Rousseau et al., 2012*). Previously it was thought that these exclusively GABAergic synapses were made by non-glycinergic GoCs. Our findings suggest that PCs likely constitute most of these synapses.

PCs inhibit UBCs by activating both GABA$_A$Rs and GABA$_B$Rs, and light-evoked GABA$_B$ currents are observed in a larger percentage of mGluR1$^+$ UBCs (>90%) than are light-evoked GABA$_A$ currents (29%). This difference in prevalence likely reflects differential sensitivities of GABA$_A$Rs and GABA$_B$Rs. Activation of GABA$_A$Rs on UBCs requires high GABA levels (*Möhler, 2006*), and thus requires a direct PC synapse for effective activation. GABA$_B$Rs are activated by much lower levels of GABA (*Galvez et al., 2000*; *Kaupmann et al., 1998*). Consequently, a direct PC input is not required and GABA released from PC synapses onto neighboring cells can pool and spillover to activate GABA$_B$Rs. This also suggests that the combined activity of many PCs could efficiently activate GABA$_B$Rs on UBCs and control spontaneous activity.

It is important to note that we have focused on PC feedback to UBCs in lobule X. Both UBCs and PC collaterals are also present throughout the cerebellar cortex, albeit at a lower density. It remains to be seen whether PC to UBC feedback occurs in other regions, and if it does, whether it has the same target dependence.

## Comparisons to properties of PC-GrC feedback

Within the input layer, PCs also directly inhibit GrCs, and there are similarities and differences in the inhibition of these two targets. The failure rate and potency of the fast component of inhibition are similar for PC synapses onto UBCs and GrCs (*Guo et al., 2016*). Therefore, we hypothesize that the overall effect of fast and stochastic GABA$_A$R PC inhibition of UBCs and GrCs is to lower the gain of the input layer. The slow GABA$_B$ feedback to UBCs occurs in concert with a slow inhibition of GrCs that is mediated by a very different mechanism (*Guo et al., 2016*). PCs slowly inhibit GrCs primarily by activating special high-affinity α6 and δ subunit-containing GABA$_A$Rs that are specialized to respond to low levels of extracellular GABA (*Brickley et al., 1996*; *Hamann et al., 2002*; *Kaneda et al., 1995*; *Wall and Usowicz, 1997*). It appears that α6/δ containing GABA$_A$Rs in GrCs and GABA$_B$Rs in UBCs can both respond to the low GABA levels that reflect the population-averaged PC firing rates, and in both cases, the kinetics of the responses are slow (312 ms in GrCs vs. 580 ms in UBCs). The major differences between slow PC inhibition of UBCs and GrCs are the state change that can occur in UBCs, and the target dependence of the UBC inhibition, neither of which have been described for PC inhibition of GrCs (*Guo et al., 2016*).

## Potential functional consequences of slow PC to UBC inhibition

Slow PC-UBC inhibition represents the average synaptic contribution from multiple PCs, and when the firing rates of these PCs are elevated UBCs are hyperpolarized. Increases in PC firing lead to slow increases in inhibition, and following decreases in PC firing rate this inhibition gradually decreases. Importantly, this can alter the state of the UBC. As has been shown previously, hyperpolarization can relieve the inactivation of T-type calcium channels and make neurons burst upon depolarization (*Perez-Reyes, 2003*). Concordantly, we observed rebound effects in some UBCs when PC firing returned to baseline (*Figure 4H*, last row and *Figure 4I*) as well as steeper initial input/output

relationship for depolarizing current steps with PC inhibition (*Figure 5E,F*). It is likely that this component also allows PCs to constitutively regulate the properties of UBCs (although we have not addressed this possibility in our brain slice experiments where many PC axons and connections to UBCs are not intact). We conclude that PC firing may be able to alter UBC-mediated transformations in the input layer of the cerebellum. Slow inhibition could also allow PCs to dynamically regulate the excitability of mGluR1$^+$ UBCs. MF inputs to the cerebellum can modulate the firing of PCs, and these firing rate changes will feedback to mGluR1$^+$ UBCs to alter their firing. This could be particularly important for UBCs that fire for more than 10 s in response to a brief MF activation in the absence of inhibition (*Guo et al., 2021*).

# Materials and methods

## Key resources table

| Reagent type (species) or resource | Designation | Source or reference | Identifiers | Additional information |
|---|---|---|---|---|
| Strain, strain background (*Mus musculus*) | C57BL/6 | Charles River | | |
| Strain, strain background (*M. musculus*) | B6.Cg-Tg(*Pcp2-cre*)3555Jdhu/J | Jackson Laboratory | Strain#010536 | |
| Strain, strain background (*M. musculus*) | ChR2-EYFP (Ai32) | Jackson Laboratory | Strain#024109 | |
| Strain, strain background (*M. musculus*) | Synaptophysin-tdTomato (Ai34D) | Jackon Laboratory | Strain#021570 | |
| Chemical compound, drug | NBQX disodium salt | Abcam | Ab120046 | |
| Chemical compound, drug | (R)-CPP | Abcam | Ab120159 | |
| Chemical compound, drug | Strychnine hydrochloride | Abcam | Ab120416 | |
| Chemical compound, drug | SR95531 (Gabazine) | Abcam | Ab120042 | |
| Chemical compound, drug | L-Glutamate | Abcam | Ab120049 | |
| Chemical compound, drug | (RS)-3,5-DHPG | Tocris | Cat. no. 0342/1 | |
| Chemical compound, drug | (R,S)-Baclofen | Abcam | Ab120149 | |
| Chemical compound, drug | CGP 55845 hydrochloride | Tocris | Cat. no. 1248 | |
| Antibodies | Guinea pig anti-VGAT (Guinea pig polyclonal) | Synaptic Systems | 131004 | 1 μg/mL stock, 1:200 |
| Antibodies | Mouse anti-rat mGluR1 (Mouse monoclonal) | BD Pharmingen | Cat. no. 556331 | 0.5 mg/mL stock, 1:800 |
| Antibodies | Goat anti-Guinea pig Alexa Fluor 488 (Goat polyclonal) | Abcam | Ab150185 | 1:500 |
| Antibodies | Goat anti-mouse Alexa Fluor 647 (Goat poly clonal) | Invitrogen | A32728 | 1:500 |
| Software, algorithm | Igor Pro 6 | Wavemetrics | https://www.wavemetrics.com/ | |

*Continued on next page*

*Continued*

| Reagent type (species) or resource | Designation | Source or reference | Identifiers | Additional information |
|---|---|---|---|---|
| Software, algorithm | MafPC | Courtesy of M.A. Xu-Friedman | https://www.xufried man.org/mafpc | |
| Software, algorithm | MATLAB (R2017a) | MathWorks | https://www.mathworks. com/products/matlab.html | |
| Software, algorithm | ImageJ | NIH | https://imagej.nih.gov/ij/ | |
| Software, algorithm | Python | Python Software Foundation | https://www.python.org/ | |
| Software, algorithm | Pytorch | Opensource | https://www.pytorch.org/ | |

## Immunohistochemistry and confocal imaging

Pcp2-Cre (Jackson Laboratory, 010536) × synaptophysin-tdTomato (Ai34D, Jackson Laboratory, 012570) P40 mice were anesthetized with intraperitoneal injections of ketamine/xylazine/acepromazine mixture at 100, 10, and 3 mg/kg, and transcardially perfused with 4% PFA in phosphate-buffered saline solution. The brain was removed from the skull and post-fixed overnight in 4% PFA. The cerebellar vermis and flocculus were dissected out and embedded in 6% low melting agarose before slicing. Sagittal vermal slices and coronal floccular slices at 50 µm were obtained from two animals using Leica VT1000S vibratome (Leica Biosystems, Buffalo Groves, IL). For immunolabeling of inhibitory synapses and the UBCs, slices were permeabilized (0.4% Triton X-100 and phosphate-buffered saline [PBS]) for 30 min and blocked for 1 hr at room temperature (0.4% Triton X-100, 4% normal goat serum) and incubated overnight at 4°C with primary antibodies (Guinea pig anti-VGAT, Synaptic Systems 131004, 1 µg/mL, 1:200 and mouse anti-mGluR1, BD Pharmingen, 0.5 mg/mL, 1:800 in 0.2% Triton X-100%, and 2% normal goat serum). The following day, the slices were washed three times for 10 min in PBS and incubated with secondary antibodies overnight at 4°C (Goat anti-Guinea pig-Alexa Fluor 488, Abcam ab150185, 1:500 and Goat anti-mouse Alexa Fluor 647, Invitrogen A32728, 1:500). Slices were mounted with a #1.5 coverslip using ProLong Diamond Antifade Mountant without DAPI (Thermo Fisher Scientific, Waltham, MA).

Z-stack images of entire vermal and floccular slices were obtained on the Olympus FV3000 confocal microscope using the Multi-Area Time-Lapse Software module. A 60× oil immersion objective with 1.42 NA was used and the x, y, z resolutions were 0.212, 0.212, and 0.906 µm, respectively. Each z-step was 1 µm and the entire stack was 20 µm in thickness.

## Image analysis

PC synapses were detected using a modified deep neural network that is based on the U-Net (*Ronneberger et al., 2015*). A total of two brains were analyzed. The network was fed in all three imaging channels including the mGluR1 labeling. The rationale being that while colocalization of synaptophysin and VGAT was sufficient for synapse detection, mGluR1 provided additional contextual information for annotation in the molecular layer as it densely labeled PC dendrites. The network was trained using soft-dice loss on 100 annotated images (a single slice in z, 50×50 µm² in width and height) and validated with an 80/20 train-test-split. Our human annotator criteria were that the inhibitory PC synapses should be synaptophysin-TdTomato and VGAT positive puncta with clear morphological features that are present across at least two to three consecutive z-slices. After training the network, we verified that it works consistently across different cerebellar layers and regions using out-of-sample validation. We found that using an ensemble of 10 independently trained networks qualitatively improved the annotation by reducing the amount of variability in synapse identification. The true positive rate of synapse detection was at 95.5% and the false-positive rate was 10.1%. We note however that the false positive rate was likely inflated as the human annotator sometimes missed difficult-to-spot synapses (*Figure 1—figure supplement 1A–B*, yellow arrows). For illustrating the UBCs, the mGluR1 signals in the molecular layer were manually cropped in the

final composite image. The neural network approach was necessary to generate high-quality overviews of synaptic density maps over large regions.

## Slice preparation for electrophysiology

Adult (P30–P40) C57BL/6 or Pcp2-Cre×ChR2-EYFP (Ai32, Jackson Laboratory, 024109) mice were first anesthetized with an intraperitoneal injection of ketamine/xylazine/acepromazine mixture at 100, 10, and 3 mg/kg, and transcardially perfused with an ice-cold choline slicing solution consisting of (in mM): 110 choline Cl, 7 $MgSO_4$, 2.5 KCl, 1.2 $NaH_2PO_4$, 0.5 $CaCl_2$, 11.6 Na-ascorbate, 2.4 Na-pyruvate, 25 $NaHCO_3$, and 25 glucose equilibrated with 95% $O_2$ and 5% $CO_2$. The cerebellum was dissected, mounted against an agar block, and submerged in the choline solution during slicing. Sagittal slices of the vermis were obtained using a Leica VT1200S vibratome and allowed to recover for 30 min at 33°C in the artificial cerebral spinal fluid (ACSF) consisting of (in mM): 125 NaCl, 26 $NaHCO_3$, 1.25 $NaH_2PO_4$, 2.5 KCl, 1 $MgCl_2$, 1.5 $CaCl_2$, and 25 glucose (pH 7.4, osmolarity 315) equilibrated with 95% $O_2$ and 5% $CO_2$. The incubation chamber was then removed from the warm water bath and kept at room temperature for recording for up to 6 hr.

## Electrophysiology

Recordings were performed at 34–36°C in ASCF containing 5 µM NBQX, 2 µM R-CPP, and 1 µM strychnine set to a flow rate of 2 mL/min. Visually guided recording of UBCs was obtained under a 60× objective with differential interference contrast imaging on an Olympus BX51WI microscope. Identity of the UBCs was also verified with fluorescent dye after recording (Alexa Fluor 594, 100 µM). Borosilicate patch pipette (3–5 MΩ) containing either a KCl internal (in mM: 140 KCl, 4 NaCl, 0.5 $CaCl_2$, 10 HEPES, 4 MgATP, 0.3 NaGTP, 5 EGTA, and 2 QX-314, pH adjusted to 7.2 with KOH) or a K-methanesulfonate internal (in mM: 122 K-methanesulfonate, 9 NaCl, 9 HEPES, 0.036 $CaCl_2$, 1.62 $MgCl_2$, 4 MgATP, 0.3 Tris-GTP, 14 Tris-creatine phosphate, and 0.18 EGTA, pH7.4) was used for whole-cell recordings of $GABA_A$R- or $GABA_B$R-mediated current, respectively. A junction potential of −8 mV was corrected for the K-Methansulfonate internal during recording. For cell-attached UBC recordings, ACSF was used as the internal solution. For perforated-patch recording, an internal containing (in mM) was used: 100 K-methanesulfonate, 13 NaCl, 2 $MgCl_2$, 10 EGTA, 1 $CaCl_2$, 10 HEPES, 0.1 Lucifer Yellow, and 0.25 amphotericin B, pH 7.2 with KOH. A junction potential of +7 mV was corrected online. For recordings of the $GABA_B$R-mediated current, 10 µM SR95531/gabazine was included in the bath to isolate the metabotropic inhibition. For paired PC-UBC recordings, a whole-cell UBC recording with KCl was obtained first. Then 10–20 PCs in lobule X were screened with cell-attached recordings using a large pipette (1–2 MΩ) containing ACSF.

## Pharmacology

For identification of UBC subtypes in either whole-cell or perforated recordings, ACSF containing glutamate (1 mM, 50 ms) or DHPG (100 µM, 20 ms) was pressure-applied at 5 psi through a borosilicate pipette with a Picospritzer III (Parker Hannifin, Hollis, NH). For the sequential mGluR1 and $GABA_B$R activation experiment (*Figure 5G–J*), concentrations and durations of pressure applications are 100 µM, 50 ms for DHPG, and 250 µM, 100 ms for baclofen. The same baclofen concentration and duration were used for testing the stability of $GABA_B$R-mediated current (*Figure 4—figure supplement 1*). For wash-in experiments, solution exchange was controlled via ValveLink8.2 Controller (AutoMate Scientific, Berkeley, CA). Gabazine (SR95531, 5 µM) was used for blocking $GABA_A$R and CGP (1 µM) was used for blocking $GABA_B$R.

## Optogenetics

Slices from Pcp2-Cre×ChR2-EYFP were kept in the dark before recording. A laser source (MBL-III-473-50 mW, Optoengine) was fiber coupled to the excitation path of the microscope. For over bouton stimulation, brief (0.5 ms) high intensity (160 mW/$mm^2$) light pulses were delivered under the 60× objective and focused down to a 50-µm spot. For the optical modulation of PC spontaneous firing, much lower intensities (10 or 25 µW/$mm^2$) of light were delivered through the 10× objective over the entire lobule X (~1 mm spot) for 2 s. To ensure the quality of the slice, we recorded many PCs before the experiment to check for spontaneous activity and to ensure that these low-intensity

stimulations did not result in PCs bursting. Only healthy slices were used for subsequent UBC recordings.

## Data acquisition and analysis

Electrophysiology experiments were performed with a MultiClamp 700B amplifier (Molecular Device) and controlled by mafPC (Matthew Xu-Friedman, SUNY Buffalo) in Igor Pro 7 (WaveMetrics). Data were filtered at 4 kHz in MultiClamp and digitized at 50 kHz by InstruTECH ITC-18 (Heka Instrument Inc). Analysis of DHPG evoked spiking in UBC was done via peak detection in MATLAB. sIPSC timing was determined by time of 5% to peak threshold crossing. The timing of the PC spike for determining the latency of synaptic transmission was determined by the first derivative of the action potential. The time constant for the slow $GABA_BR$-mediated current was obtained using a single exponential fit. Data are reported as mean ± SEM.

## Acknowledgements

Imaging was performed in the Vision Core and NINDS P30 Core Center (NS072030) to the Neurobiology Imaging Center at Harvard Medical School with the help of M El-Rifai (performed RNA scope) and M Ocana. The authors thank Laurens Witter for his insights in the early stages of this project, and Vincent Huson for feedback on the manuscript.

## Additional information

### Funding

| Funder | Grant reference number | Author |
|---|---|---|
| National Institute of Neurological Disorders and Stroke | R35NS097284 | Wade G Regehr |
| Stuart H.Q. & Victoria Quan Fellowship | | Chong Guo |

The funders had no role in study design, data collection and interpretation, or the decision to submit the work for publication.

### Author contributions

Chong Guo, Conceptualization, Data curation, Software, Formal analysis, Investigation, Visualization, Writing - original draft, Writing - review and editing; Stephanie Rudolph, Investigation, Writing - review and editing; Morgan E Neuwirth, Software, Investigation; Wade G Regehr, Conceptualization, Resources, Funding acquisition, Writing - original draft, Writing - review and editing

### Author ORCIDs

Chong Guo (iD) https://orcid.org/0000-0002-1230-5333
Stephanie Rudolph (iD) https://orcid.org/0000-0003-0388-7762
Wade G Regehr (iD) https://orcid.org/0000-0002-3485-8094

### Ethics

Animal experimentation: Animal experimentation: All experiments were conducted in accordance with federal guidelines and protocols (#1493) approved by the Harvard Medical Area Standing Committee on Animals.

### Decision letter and Author response

Decision letter https://doi.org/10.7554/eLife.68802.sa1
Author response https://doi.org/10.7554/eLife.68802.sa2

## Additional files

### Supplementary files

• Transparent reporting form

### Data availability

The RNAseq data for cerebellar UBCs can be visualized through (https://singlecell.broadinstitute.org/single_cell/study/SCP795/). Raw and processed data that support the findings of this study have been deposited in GEO under accession number GSE165371 and in at the Neuroscience Multi-omics (NeMO) Archive (https://nemoarchive.org/).

The following datasets were generated:

| Author(s) | Year | Dataset title | Dataset URL | Database and Identifier |
|---|---|---|---|---|
| Guo C, Rudolph S | 2021 | Purkinje cell outputs selectively inhibit a subset of unipolar brush cells in the input layer of the cerebellar cortex | https://github.com/chongguo/PC-UBC_elife2021 | GitHub, PC-UBC_elife2021 |
| Kozareva V, Martin C, Osorno T, Rudolph S, Guo C, Vanderburg C, Nadaf N, Regev A, Regehr W, Macosko E | 2021 | A transcriptomic atlas of mouse cerebellar cortex reveals novel cell types | https://www.ncbi.nlm.nih.gov/geo/query/acc.cgi?acc=GSE165371 | NCBI Gene Expression Omnibus, GSE165371 |

The following previously published dataset was used:

| Author(s) | Year | Dataset title | Dataset URL | Database and Identifier |
|---|---|---|---|---|
| Kozareva V, Martin C, Osorno T, Rudolph S, Guo C, Vanderburg C, Nadaf N, Regev A, Regehr W, Macosko E | 2021 | A transcriptomic atlas of mouse cerebellar cortex reveals novel cell types | https://singlecell.broadinstitute.org/single_cell/study/SCP795/a-transcriptomic-atlas-of-the-mouse-cerebellum | Single Cell Portal, SCP795 |

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
