## [Decision Letter]

**Acceptance summary:**

This paper expands our understanding of the cerebellar microarchitecture, and, in particular, the level of recurrence in this circuit. A combination of electrophysiological, pharmacological and optogenetic approaches demonstrate a new inhibitory feedback connection from Purkinje cells onto a subset of unipolar brush cells. This is an important discovery for the cerebellar field, and should also be of interest to readers studying feedback in neural circuits.

**Decision letter after peer review:**

Thank you for submitting your article "Purkinje cell outputs selectively inhibit a subset of unipolar brush cells in the input layer of the cerebellar cortex" for consideration by *eLife*. Your article has been reviewed by 3 peer reviewers, including Jennifer L Raymond as the Reviewing Editor and Reviewer #1, and the evaluation has been overseen by Kenton Swartz as the Senior Editor. The following individual involved in review of your submission has agreed to reveal their identity: Philippe Isope (Reviewer #2).

Essential revisions:

The finding that PCs inhibit mGluR1 positive UBCs is important in its own right and well established by the evidence in Figure 2 and 4.

1) The finding of feedback from the PCs to the UBCs opens up exciting new research directions. The reviewers appreciated the authors' efforts to gain insight about the signal processing function of this new feedback connection, however, they agreed that the specific claims about the function of the fast and slow inhibitory feedback were not well supported by the evidence provided, and hence should be clearly identified as highly speculative and the conclusions significantly tempered. Additional evidence to support the claims about the specific signal processing functions of the fast and slow feedback would be highly desirable, but is not essential. See specific suggestions below.

2) Some clarifications are needed regarding the histology:

– What is the reader supposed to take away from the convolutional neural network analysis of Purkinje cell synapses onto UBCs? It seems that two brains were annotated by a human expert, and then the network was trained on those two brains and could reproduce similar results on held-out data, as shown in Figure 1-Supp 1. What criteria did the human annotator use in marking synapses? Was the neural network approach used for anything beyond what the human annotator did?

– Is it absolutely certain that there is no mGluR1 expression in Lugaro cells and Golgi cells? (See Knöpfel and Grandes, 2002 and Grandes et al. 1994.) The high magnification images in Figure 1K do show the characteristic morphology of UBCs, but there is a possibility that low mag. images of distribution in lobules could include these other cell types in spite of the difference in size, in case they express mGluR1. This possibility could change the interpretation of these results, so please clarify.

– There seem to be PC synapses on the cell body of UBCs as well as on the brush (Figure 1K), although the synapses on the brush are highlighted for mention. Please clarify.

Specific, non-essential suggestions for the authors to consider, should they decide to strengthen the evidence about the function of the fast and slow feedback.

– There is the implicit assumption that regular PC firing hardly affects mGluR1 positive UBC discharge, but it would be interesting to address or at least discuss constitutive PC inhibition. For example, this could be assessed in UBCs that don't express mGluR2, by applying an mGluR2 agonist to inhibit Golgi cell discharge, then GABA_A_+GABA_B_ blockers in the bath.

– Tonic UBC hyperpolarization by steady PC excitation in figure 5D-F was not fully quantified. For example, is there different behaviors at different light intensities as in Figure 4? In figure 5D-F, a bigger burst at the beginning of the depolarization step was observed, but the steady state current seems unchanged. This is an important point, as rebound discharge is an important physiological feature of UBCs (Diana MA et al., J Neurosci. 2007) which may modify the current knowledge about ON and OFF UBC computation. Noticeably, there is a clear rebound discharge in Figure 4I but not quantified. The interplay between T-type channel and GABA_B_ hyperpolarization may be addressed or discussed more.

– Is there any combined effect of GABA_A_ and GABA_B_ signaling during trains of stimulation? Probability of release at the PC-UBC connection appeared low, but we expect a buildup of GABA_A_ mediated current during trains and a synergistic effect with GABA_B_ signaling on UBC firing rate. However, in the current form, experiments were performed either with GABA_A_ or GABA_B_ antagonist in the bath. Experiments with neither GABA_A_ nor GABA_B_ blockers would likely be informative.

– Can you draw any conclusions on the number of PC inputs to a UBC from the paired recordings? How many PCs are required to inhibit UBCs via GABA_A_ or GABA_B_ receptors? This question is important as groups of PCs can be synchronized or PCs can fire in burst at high frequencies, suggesting that the low probability of release of GABA may act as a high pass filter for GABA_A_ signaling. While paired recordings (Figure 3) are challenging, loose-cell attached recordings on PCs (Barbour and Isope, J Neurosci. Methods, 2000) or focalized light illumination in ChR2 expressing PCs maybe used instead of cell-attached.

– The study focused on mGluR1 positive cells, but showed that Gabbr1 is also expressed in other UBCs. Therefore, GABA_B_ inhibition could be relevant for mGluR1 negative UBCs as well. If the authors have these data, it would be an interesting extension.

– A discussion on the physiological consequences of a selective inhibition of mGluR1 positive vs mGluR2 UBCs would strengthen the study. Can we extrapolate that "ON" UBCs are inhibited by PCs while "OFF" cells are not?

– Quantification of the GABA_A_ current recorded in figure 2 would be informative as it relates to glutamate induced current or DHPG induced increase in firing rate.

– It has been thought that UBCs serve to prolong GrC firing after brief mossy fiber inputs. The negative feedback described here would serve the opposite purpose. Thus, it is important to understand the time course of the negative feedback relative to mossy fiber input.

– The major conclusions of this paper relate to feedback regulation of the input layer. But the data presented here don't directly show a feedback loop, i.e., that the GrCs that feedforward mossy fiber information to the Purkinje cells are the ones that are regulated by the UBCs that these PCs project to. Given the observation of slow GABA_B_-dependent feedback, the role of direct feedback vs. feedback onto UBCs without a direct connection is important.

– In relation to the previous question, it remains unclear how modulating UBC input would affect Purkinje cells.

– The authors previously showed that PC-GrC feedback is also heterogeneously distributed, and will serve to inhibit the input layer. Since PCs directly provide both fast and slow feedback to GrCs in these regions, the PC-UBC connection demonstrated here suggests additional negative feedback via UBCs to GrCs. Therefore, the effectiveness of second level of negative feedback regulation of PCs to the input layer remains unclear.

– Until the current finding of PC-UBC feedback, inhibitory inputs to UBCs were thought to be limited to those from Golgi cells. The relative time course of Golgi cell vs. PC feedback inhibition in controlling UBCs is worth consideration.

– It would be important to know whether PC feedback to UBCs is in the same parasagittal plane.

– As an additional example that relates to the same overall question about how this feedback loop will actually affect neural signals via mossy fibers, the authors also show that the faster inhibitory mGluR2/3-mediated current comes first, and the mGluR1-mediated excitation comes later (Guo et al., Biorxiv, 2020). When would PC feedback come, in relation to this sequence?

– There is known to be a heterogeneous distribution of UBCs, as shown in this study too. Is there also preferential localization of mGluR1-positive UBCs? Figure 1 shows only mGluR1-positive UBCs. Since the presence of mGluR1 and the response to glutamate are related to PC feedback, it is important to know whether the proportion of UBCs that do not get PC feedback even in these UBC-rich regions is different from elsewhere, or not. This would have important implications for circuit processing in vestibular-input-related vs. other regions

– A significant minority of DHPG-responsive cells don't have PC feedback (although the absence of an optogenetically-induced response is not definitive proof of lack of PC inputs). Notwithstanding this caveat, I think this significant minority should also be discussed.

– In the PC-UBC paired recordings, were there UBC responses that were not timed to PC spikes?

– The entire study is motivated by describing specialization of different regions of the cerebellum, but the physiology was only performed in lobule X of vermis. Although there are fewer UBCs in other parts of the cerebellum, do the same patterns of feedback from the Purkinje cells apply?

Summary:

The major strength of this study is the straightforward and convincing demonstration of a previously unknown synaptic connection from Purkinje cells (PCs) to mGluR1-positive unipolar brush cells (UBCs) in lobule X of the cerebellum. UBCs are interneurons of the cerebellar input layer, which preferentially distributed in some lobules of the cerebellum; especially those involved in vestibular function. The authors utilize a combination of electrophysiological recordings from UBCs in the acute rodent cerebellar slice, optogenetic activation of PCs, and pharmacological manipulations to establish that PCs inhibit UBCs in cerebellar lobule X through both fast synaptic transmission through GABA_A_ receptors, as well as long-lasting inhibition via GABA_B_ receptors. Since PCs constitute the output stage of the cerebellar cortex, the inhibitory signaling described in this study identifies a new feedback control pathway to the input layer, which constitutes a computational specialization of lobule X.

Another strength of this study is the determination of how molecular heterogeneity corresponds to circuit connectivity and variable responses to synaptic input. There is heterogeneity in the molecular profiles of UBCs, and expression of mGluR1 is a key marker for a sub-population of UBCs. The authors demonstrate that mGluR1-expressing UBCs are preferentially targeted by PC feedback. – It is noteworthy that GABA_A_-mediated PC feedback affects a majority of, but not all, mGluR1-containing UBCs, but GAABB-mediated PC feedback affected all UBCs tested. This finding suggests tonic control of the UBC by PCs even without a direct PC connection.

The results raise the interesting question of the signal processing and computational function of this newly identified inhibitory feedback pathway, which should inspire future research. The authors offer some ideas along with limited evidence, however, the reviewers felt that additional research would be required to establish the authors' claims about the specific roles of PC inhibition of UBCs via GABA_A_ and via GABA_B_, especially in the context of the full complement of signaling pathways in the intact cerebellar circuit.

The claim that "phasic GABA_A_R-mediated PC-UBC synapses convey noisy and unreliable

output to input layer feedback" is based on recordings from two pairs of synaptically recorded PCs and UBCs. The reviewers fully appreciate how technically challenging these experiments are, nevertheless it is difficult to conclude much from n=2 observations. Moreover, even if the high (85%) synaptic failure rate were to be replicated in a large number of PC-UBC pairs, it is not clear that this should be interpreted as "noise." Purkinje cells fire at high rates of 50-100 Hz, and one would expect a buildup of GABA_A_ mediated current during spike trains and potentially a synergistic effect with GABA_B_ signaling on UBC firing rate.

Likewise, the pharmacological approach used in Figure 5G-J lacks the temporal resolution to establish the claim that "Slow GABA_B_R-mediated inhibition allows elevated PC activity to sharpen the input-output transformation of UBCs."

[Editors' note: further revisions were suggested prior to acceptance, as described below.]

Thank you for resubmitting your work entitled "Purkinje cell outputs selectively inhibit a subset of unipolar brush cells in the input layer of the cerebellar cortex" for further consideration by *eLife*. Your revised article has been evaluated by Kenton Swartz as the Senior Editor, and a Reviewing Editor.

The two main conclusions from the original review still apply in their entirety:

1) The finding that PCs inhibit mGluR1 positive UBCs is important in its own right and well established by the evidence provided.

2) The specific claims about the function of the fast and slow inhibitory feedback were not well supported, in particular the following two claims, which are made in the last two lines of the abstract "GABA_A_R-mediated feedback is fast, unreliable, noisy, and may help linearize input-output curves and decrease gain. Slow GABA_B_R-mediated inhibition could allow elevated PC activity to sharpen the input-output transformation of UBCs, and allow dynamic inhibitory feedback of mGluR1-expressing UBCs."

During the review of the original submission, the reviewers and editor agreed there were two reasonable paths forward to publication: i) provide more evidence for the unsubstantiated claims, or ii) simply back off on the unsubstantiated claims, and focus on the main finding of the new connection from PCs to mGluR1+ UBCs. Inexplicably, the authors did not pursue either of these options, but rather resubmitted the original manuscript almost verbatim, complete with the two unsubstantiated claims at the end of the abstract.

The editor apologizes if placement of the key concerns of the reviewers at the end of the decision letter caused the authors to overlook these central concerns, which left the reviewers completely unconvinced of the functional claims:

The claim that "phasic GABA_A_R-mediated PC-UBC synapses convey noisy and unreliable

output to input layer feedback" is based on recordings from two pairs of synaptically recorded PCs and UBCs. The reviewers fully appreciate how technically challenging these experiments are, nevertheless it is difficult to conclude much from n=2 observations. Moreover, even if the high (85%) synaptic failure rate were to be replicated in a large number of PC-UBC pairs, it is not clear that this should be interpreted as "noise." Purkinje cells fire at high rates of 50-100 Hz, and one would expect a buildup of GABA_A_ mediated current during spike trains and potentially a synergistic effect with GABA_B_ signaling on UBC firing rate.

Likewise, the pharmacological approach used in Figure 5G-J lacks the temporal resolution to establish the claim that "Slow GABA_B_R-mediated inhibition allows elevated PC activity to sharpen the input-output transformation of UBCs."

The authors provided a lengthy rebuttal of the suggestions the reviewers had offered about how the unsubstantiated claims might be strengthened, however, the second option of removing those claims still seems like a reasonable path forward, if the authors are willing to substantially revise the manuscript.

---

## [Author Response]

Essential revisions:The finding that PCs inhibit mGluR1 positive UBCs is important in its own right and well established by the evidence in Figure 2 and 4.1) The finding of feedback from the PCs to the UBCs opens up exciting new research directions. The reviewers appreciated the authors' efforts to gain insight about the signal processing function of this new feedback connection, however, they agreed that the specific claims about the function of the fast and slow inhibitory feedback were not well supported by the evidence provided, and hence should be clearly identified as highly speculative and the conclusions significantly tempered. Additional evidence to support the claims about the specific signal processing functions of the fast and slow feedback would be highly desirable, but is not essential. See specific suggestions below.

We have addressed the specific issues raised below.

2) Some clarifications are needed regarding the histology:– What is the reader supposed to take away from the convolutional neural network analysis of Purkinje cell synapses onto UBCs? It seems that two brains were annotated by a human expert, and then the network was trained on those two brains and could reproduce similar results on held-out data, as shown in Figure 1-Supp 1. What criteria did the human annotator use in marking synapses? Was the neural network approach used for anything beyond what the human annotator did?

We have clarified the explanation in both the results and methods section. We emphasize that we did not manually annotate two brains. The expert annotator only worked on 100 images, each a single 2D slice of 50x50 micron. This is less than 0.001% of the parasagittal vermis volume in Figure 1C for reference. Our human annotator criteria were that the inhibitory PC synapses should be synaptophysin-TdTomato and VGAT positive puncta with clear morphological features that are present across at least two to three consecutive z-slices. After training the network, we verified that it works consistently across different cerebellar layers and regions using out of sample validation. The neural network approach was necessary to generate high-quality overviews of synaptic density maps over large regions. (pages 12-13).

– Is it absolutely certain that there is no mGluR1 expression in Lugaro cells and Golgi cells? (See Knöpfel and Grandes, 2002 and Grandes et al. 1994.) The high magnification images in Figure 1K do show the characteristic morphology of UBCs, but there is a possibility that low mag. images of distribution in lobules could include these other cell types in spite of the difference in size, in case they express mGluR1. This possibility could change the interpretation of these results, so please clarify.

The brush expresses extremely high level of mGluR1 and dominates the signals in both the high and low magnification image of UBCs. mGluR1 labeling is used in many previous papers to identify UBCs (i.e. Borges-Merjane and Trussell, 2015, Nuzi et al., 2002, McDonough et al., 2020). Please note that the entire image was obtained at high resolution, which allows us to inspect the morphology of the labelled structures. The distinctive, intense mGluR1 immunofluorescence pattern we observe is characteristic of UBC brush labelling, and is not consistent with Golgi cell or Lugaro cell labelling. (page 4)

– There seem to be PC synapses on the cell body of UBCs as well as on the brush (Figure 1K), although the synapses on the brush are highlighted for mention. Please clarify.

We agree that synapses are near both the brush and the cell body in these images, and we have modified the legend and text (page 4).

Specific, non-essential suggestions for the authors to consider, should they decide to strengthen the evidence about the function of the fast and slow feedback.– There is the implicit assumption that regular PC firing hardly affects mGluR1 positive UBC discharge, but it would be interesting to address or at least discuss constitutive PC inhibition. For example, this could be assessed in UBCs that don't express mGluR2, by applying an mGluR2 agonist to inhibit Golgi cell discharge, then GABA_A_+GABA_B_ blockers in the bath.

Constitutive PC inhibition of UBCs is an interesting possibility, and we have added to the discussion to deal with this issue. UBCs can receive GABAR-mediated inhibition from GoCs and PCs. However, it is easy to sever the extensive PC collaterals that extend across lobule X, and GoCs often die in adult slices. In addition, a large percentage of UBCs express mGluR2/3Rs, so the application of an agonist is difficult to interpret. These technical challenges make it difficult to determine the magnitude and fractional contributions of constitutive PC and GoC inhibition onto UBCs.(page 10)

– Tonic UBC hyperpolarization by steady PC excitation in figure 5D-F was not fully quantified. For example, is there different behaviors at different light intensities as in Figure 4? In figure 5D-F, a bigger burst at the beginning of the depolarization step was observed, but the steady state current seems unchanged. This is an important point, as rebound discharge is an important physiological feature of UBCs (Diana MA et al., J Neurosci. 2007) which may modify the current knowledge about ON and OFF UBC computation. Noticeably, there is a clear rebound discharge in Figure 4I but not quantified. The interplay between T-type channel and GABAB hyperpolarization may be addressed or discussed more.

The reviewer makes an important point. We now point out the rebound discharge in Figure 4 and discuss in the context of T-type calcium channels and the study of Diana et al. (2007). We discuss how pauses in PC firing could potentially generate rebound firing in UBCs. (page 10)

– Is there any combined effect of GABA_A_ and GABA_B_ signaling during trains of stimulation? Probability of release at the PC-UBC connection appeared low, but we expect a buildup of GABA_A_ mediated current during trains and a synergistic effect with GABA_B_ signaling on UBC firing rate. However, in the current form, experiments were performed either with GABA_A_ or GABA_B_ antagonist in the bath. Experiments with neither GABA_A_ nor GABA_B_ blockers would likely be informative.

We had also considered the possibility of performing experiments in the absence of any GABA receptor antagonists. Ultimately, we felt that it was more informative to explore the contributions of GABA_A_ and GABA_B_ receptors in isolation, and that performing experiments in the absence of GABAR antagonists did not provide a sufficient additional advance to justify the considerable time and effort required.

– Can you draw any conclusions on the number of PC inputs to a UBC from the paired recordings? How many PCs are required to inhibit UBCs via GABA_A_ or GABA_B_ receptors? This question is important as groups of PCs can be synchronized or PCs can fire in burst at high frequencies, suggesting that the low probability of release of GABA may act as a high pass filter for GABA_A_ signaling. While paired recordings (Figure 3) are challenging, loose-cell attached recordings on PCs (Barbour and Isope, J Neurosci. Methods, 2000) or focalized light illumination in ChR2 expressing PCs maybe used instead of cell-attached.

We share the reviewer’s desire to determine the number of PCs that can influence each UBC. However, we do not think that electrophysiological studies will provide a reliable estimate of the number of PCs that inhibit a UBC. This is essentially two questions, because it is likely that direct contacts are required for GABA_A_ responses, GABA_B_ responses do not require direct contacts and can be activated by spillover. In order to estimate input numbers, it is necessary to determine the average size of an individual input and determine the size of the response to all inputs activated simultaneously. We used loose-cell attached recordings to record from PCs, but these experiments were very challenging and the n’s were small (Figure 3). We are less confident about our ability to reliably activate all PC inputs to a UBC. There is the challenge that over bouton stimulation and the use of ChR2 can artificially elevate the initial probability of release. We also think it likely that many PC axons and their associated synapses may be damaged, which would cause us to underestimate the number of inputs.

For these reasons, in future studies (within the next 2 or 3 years) we hope to determine the total number of PC inputs onto a single UBC using serial EM reconstruction, which we feel will provide much more reliable estimates. Estimating the number of PCs that can contribute to the GABA_B_ response is much more challenging because in addition to the challenges outlines for direct GABA_A_ synapses, it is extremely difficult to determine the magnitude of a single PC to UBC GABA_B_ component. It is also highly likely that responses from different inputs will not sum linearly and that there will be a strong frequency dependence to the magnitude. For these reasons we prefer to be cautious and not draw conclusions that go beyond our data regarding the numbers of inputs.

– The study focused on mGluR1 positive cells, but showed that Gabbr1 is also expressed in other UBCs. Therefore, GABA_B_ inhibition could be relevant for mGluR1 negative UBCs as well. If the authors have these data, it would be an interesting extension.

It is generally thought that GABA_B_ receptors are heterodimers consisting of GABA_B1_ and GABA_B2_ receptors (Frangaj and Fan 2018). It is therefore of considerable interest that *Gabbr2* shows such a pronounced gradient and such low expression levels in mGluR1- cells, but Gabbr1 expression does not exhibit such a gradient. We now comment on this in the text (page 6)

– A discussion on the physiological consequences of a selective inhibition of mGluR1 positive vs mGluR2 UBCs would strengthen the study. Can we extrapolate that "ON" UBCs are inhibited by PCs while "OFF" cells are not?

We have added additional discussion in the possible physiological consequences of PCs selectively inhibiting a subset of UBCs. In this discussion we strive to avoid the convenient “ON: and “OFF” terms that we feel are insufficiently precise. We have demonstrated in a recent manuscript (Guo et al. 2021, Nat Comm in press, and bioRxiv 2021) that the population of UBCs have graded responses, and mGluR1 excitation is prominent in some cells, mGluR2/3 inhibition is prominent in others, but for most mGluR1 and mGluR2/3 are both present. We find that PCs preferentially inhibit UBCs with high mGluR1 expression, and predominantly excitatory MF responses.

– Quantification of the GABA_A_ current recorded in figure 2 would be informative as it relates to glutamate induced current or DHPG induced increase in firing rate.

We have provided an additional scatter plot (Figure 2N) to address this issue.

– It has been thought that UBCs serve to prolong GrC firing after brief mossy fiber inputs. The negative feedback described here would serve the opposite purpose. Thus, it is important to understand the time course of the negative feedback relative to mossy fiber input.

This is not a simple issue. In general, elevated PC inhibition will rapidly suppress firing in UBCs if there is a GABA_A_ component, and will slowly inhibit UBCs if there is a GABA_B_ component. MFs can excite many UBCs for a few hundred milliseconds to tens of seconds, and MFs inhibit many UBCs for hundreds of milliseconds by activating mGluR2/3 receptors. However, it is not clear whether MF evoked population activity in UBCs will increase or decrease PC firing, because UBCs excite granule cells that both directly excite PCs and disynaptically inhibit PCs by activating MLIs. We therefore do not know if the UBCgRCPCUBC loop leads to positive or negative feedback. Hopefully, future in vivo studies will address this issue, but they will be challenging. We feel that it is reasonable and appropriate to restrict this study and the discussion to a basic description of the PC to UBC synapse, and that many of the interesting issues raised by the reviewer are perfect topics for future studies.

– The major conclusions of this paper relate to feedback regulation of the input layer. But the data presented here don't directly show a feedback loop, i.e., that the GrCs that feedforward mossy fiber information to the Purkinje cells are the ones that are regulated by the UBCs that these PCs project to. Given the observation of slow GABAB-dependent feedback, the role of direct feedback vs. feedback onto UBCs without a direct connection is important.

We feel that it is appropriate to indicate that PCs are providing feedback from the output layer to UBCs in the input layer. Also note that each UBC excites many granule cells, and each granule cell excites about half of the local PCs. While it seems extremely likely that many GrCs activated by a UBC will directly excite the PCs that inhibit that UBC, we have not explicitly shown that this is the case.

– In relation to the previous question, it remains unclear how modulating UBC input would affect Purkinje cells.

We echo the sentiment of the reviewer. Indeed, in general the manner in which UBCs influence PC firing is complex and poorly understood, as is the case for extrinsic MFs. UBCs and MFs activate granule cells that in turn excite PCs directly and inhibit PCs through disynaptic inhibition, so the net effect is difficult to predict. We have therefore focused additional discussion on the fact that PC-UBC feedback allows the output of the cerebellar cortex to shape the temporal representation of the input layer.

– The authors previously showed that PC-GrC feedback is also heterogeneously distributed, and will serve to inhibit the input layer. Since PCs directly provide both fast and slow feedback to GrCs in these regions, the PC-UBC connection demonstrated here suggests additional negative feedback via UBCs to GrCs. Therefore, the effectiveness of second level of negative feedback regulation of PCs to the input layer remains unclear.

On page 9 we point out: The major differences in slow inhibition of UBCs and GrCs are the state change of UBCs and the target dependence of the UBC inhibition, neither of which have been described for PC inhibition of GrCs (Guo et al., 2016).

– Until the current finding of PC-UBC feedback, inhibitory inputs to UBCs were thought to be limited to those from Golgi cells. The relative time course of Golgi cell vs. PC feedback inhibition in controlling UBCs is worth consideration.

This is an interesting issue that is part of the larger question of the distinctive functional roles of PC and GC inhibition of UBCs. However, the relative times courses of GC and PC inhibition would begin with determining the activity patterns of these two cell types during behaviors.

– It would be important to know whether PC feedback to UBCs is in the same parasagittal plane.

We reconstructed PC collaterals in a previous paper across various cerebellar regions. The PC feedback appears to be confined in the parasagittal plane (Witter et al., 2016) as we now point out. This will potentially allow the PC-UBC feedback to operate within functional macrozones. (page 2)

– As an additional example that relates to the same overall question about how this feedback loop will actually affect neural signals via mossy fibers, the authors also show that the faster inhibitory mGluR2/3-mediated current comes first, and the mGluR1-mediated excitation comes later (Guo et al., Biorxiv, 2020). When would PC feedback come, in relation to this sequence?

The reviewer raises an interesting question regarding the relative timing of mGluR2 inhibition and PC inhibition. The timing of mGluR2/3 current is diverse across UBCs, ranging from about one hundred milliseconds to a few seconds. There are so many factors contributing to relative time courses of PC inhibition and mGluR2-mediated inhibition (including stimulus, MF activity pattern, continuous UBC temporal variability, the magnitudes of the GABA_A_ and GABA_B_ components, etc.) that we feel it would be premature to discuss this issue (regardless of how interesting it is).

– There is known to be a heterogeneous distribution of UBCs, as shown in this study too. Is there also preferential localization of mGluR1-positive UBCs? Figure 1 shows only mGluR1-positive UBCs. Since the presence of mGluR1 and the response to glutamate are related to PC feedback, it is important to know whether the proportion of UBCs that do not get PC feedback even in these UBC-rich regions is different from elsewhere, or not. This would have important implications for circuit processing in vestibular-input-related vs. other regions

We are interested in investigating the properties of PC to UBC inhibition in other regions of the cerebellar cortex. However, it would be necessary to essentially repeat our study in each region we wish to evaluate this issue functionally. We feel that this is beyond the scope of the current study. We deal with this issue on page 8.

– A significant minority of DHPG-responsive cells don't have PC feedback (although the absence of an optogenetically-induced response is not definitive proof of lack of PC inputs). Notwithstanding this caveat, I think this significant minority should also be discussed.

The reviewer rightly points out the very important caveat that “the absence of an optogenetically-induced response is not definitive proof of lack of PC inputs”. We have added a sentence dealing with this issue (page 6)

– In the PC-UBC paired recordings, were there UBC responses that were not timed to PC spikes?

Yes, there were IPSCs not time locked to PC spikes. This could come from either other PCs or GoCs. We did not use mGluR2 agonist to suppress GoC input in these recordings because many UBCs express mGluR2 receptors (including many mGluR1+ UBCs).

– The entire study is motivated by describing specialization of different regions of the cerebellum, but the physiology was only performed in lobule X of vermis. Although there are fewer UBCs in other parts of the cerebellum, do the same patterns of feedback from the Purkinje cells apply?

This is an interesting question that we are unable to answer definitively. Images of other regions such as lobule IX, the paraflocculus and the flocculus (Figure 1) suggest that collateral synapses may target UBCs in these regions as well. We are, however, reluctant to draw conclusions based solely on these light level images. It would require a great deal of effort to address the issue of PC inhibition of UBCs in other regions. We would need to repeat our experiments of Figures 2 and 4 for each of the other regions under consideration. We now briefly discuss this issue (page 9).

[Editors' note: further revisions were suggested prior to acceptance, as described below.]

The two main conclusions from the original review still apply in their entirety:1) The finding that PCs inhibit mGluR1 positive UBCs is important in its own right and well established by the evidence provided.2) The specific claims about the function of the fast and slow inhibitory feedback were not well supported, in particular the following two claims, which are made in the last two lines of the abstract "GABA_A_R-mediated feedback is fast, unreliable, noisy, and may help linearize input-output curves and decrease gain. Slow GABA_B_R-mediated inhibition could allow elevated PC activity to sharpen the input-output transformation of UBCs, and allow dynamic inhibitory feedback of mGluR1-expressing UBCs."During the review of the original submission, the reviewers and editor agreed there were two reasonable paths forward to publication: i) provide more evidence for the unsubstantiated claims, or ii) simply back off on the unsubstantiated claims, and focus on the main finding of the new connection from PCs to mGluR1+ UBCs. Inexplicably, the authors did not pursue either of these options, but rather resubmitted the original manuscript almost verbatim, complete with the two unsubstantiated claims at the end of the abstract.The editor apologizes if placement of the key concerns of the reviewers at the end of the decision letter caused the authors to overlook these central concerns, which left the reviewers completely unconvinced of the functional claims:The claim that "phasic GABA_A_R-mediated PC-UBC synapses convey noisy and unreliableoutput to input layer feedback" is based on recordings from two pairs of synaptically recorded PCs and UBCs. The reviewers fully appreciate how technically challenging these experiments are, nevertheless it is difficult to conclude much from n=2 observations. Moreover, even if the high (85%) synaptic failure rate were to be replicated in a large number of PC-UBC pairs, it is not clear that this should be interpreted as "noise." Purkinje cells fire at high rates of 50-100 Hz, and one would expect a buildup of GABA_A_ mediated current during spike trains and potentially a synergistic effect with GABA_B_ signaling on UBC firing rate.Likewise, the pharmacological approach used in Figure 5G-J lacks the temporal resolution to establish the claim that "Slow GABAB_R_-mediated inhibition allows elevated PC activity to sharpen the input-output transformation of UBCs."The authors provided a lengthy rebuttal of the suggestions the reviewers had offered about how the unsubstantiated claims might be strengthened, however, the second option of removing those claims still seems like a reasonable path forward, if the authors are willing to substantially revise the manuscript.

Your additional clarification has been most helpful. We now have a much better idea of the issues and the remedies. We have made a real effort to remove anything that you would consider to be speculative from the abstract, the end of the introduction and the results. We have also eliminated the discussion of the GABA_A_ component’s functional role, as per your suggestion.

The one area where I would like to push back a bit concerns Figure 5. This is the figure where we try to provide insight into what this feedback might be doing to PCs. Dr. Raymond has expressed reservations about this figure. Nonetheless, we feel that this figure adds to the paper and furthers our understanding of what PC to UBC feedback might be doing. In our point-by-point response we explain why we feel that these experiments and the associated discussion should be included. Of course, if you still feel that including Figure 5 and the associated discussion is a deal breaker, we will remove it. But we request that you allow us to retain it, because we feel that it makes the paper better.